# G4SATBench: Benchmarking and Advancing SAT Solving with Graph Neural Networks

**Zhaoyu Li**[1,2], **Jinpei Guo**[4], **Xujie Si**[1,2,3]
[1]University of Toronto, [2]Vector Institute, [3]Mila, [4]Shanghai Jiao Tong Univeristy
{zhaoyu, six}@cs.toronto.edu, mike0728@sjtu.edu.cn

## Abstract

Graph neural networks (GNNs) have recently emerged as a promising approach for solving the Boolean Satisfiability Problem (SAT), offering potential alternatives to traditional backtracking or local search SAT solvers. However, despite the growing volume of literature in this field, there remains a notable absence of a unified dataset and a fair benchmark to evaluate and compare existing approaches. To address this crucial gap, we present G4SATBench, the first benchmark study that establishes a comprehensive evaluation framework for GNN-based SAT solvers. In G4SATBench, we meticulously curate a large and diverse set of SAT datasets comprising 7 problems with 3 difficulty levels and benchmark a broad range of GNN models across various prediction tasks, training objectives, and inference algorithms. To explore the learning abilities and comprehend the strengths and limitations of GNN-based SAT solvers, we also compare their solving processes with the heuristics in search-based SAT solvers. Our empirical results provide valuable insights into the performance of GNN-based SAT solvers and further suggest that existing GNN models can effectively learn a solving strategy akin to greedy local search but struggle to learn backtracking search in the latent space.

## 1 Introduction

The Boolean Satisfiability Problem (SAT) is a crucial problem at the nexus of computer science, logic, and operations research, which has garnered significant attention over the past five decades. To solve SAT instances efficiently, modern SAT solvers have been developed with backtracking (especially with conflict-driven clause learning, a.k.a. CDCL) or local search (LS) heuristics that effectively exploit the instance's structure and traverse its vast search space [4]. However, designing such heuristics remains a highly non-trivial and time-consuming task, with a lack of significant improvement in recent years. Conversely, the recent rapid advances in graph neural networks (GNNs) [23, 27, 41] have shown impressive performances in analyzing structured data, offering a promising opportunity to enhance or even replace modern SAT solvers. As such, there have been massive efforts to leverage GNNs to solve SAT over the last few years [16, 19].

Despite the recent progress, the question of *how (well) GNNs can solve SAT* remains unanswered. One of the main reasons for this is the variety of learning objectives and usage scenarios employed in existing work, making it difficult to evaluate different methods in a fair and comprehensive manner. For example, NeuroSAT [34] predicts satisfiability, QuerySAT [30] constructs a satisfying assignment, NeuroCore [33] classifies unsat-core variables, and NSNet [28] predicts marginal distributions of all satisfying solutions to solve the SAT problem. Moreover, most previous research has experimented on different datasets that vary in a range of settings (e.g., data distribution, instance size, and dataset size),

Submitted to the 37th Conference on Neural Information Processing Systems (NeurIPS 2023) Track on Datasets and Benchmarks. Do not distribute.

which leads to a lack of unified and standardized datasets for training and evaluation. Additionally, some work [2, 35, 42] has noted the difficulty of re-implementing prior approaches as baselines, rendering it arduous to draw consistent conclusions about the performance of peer approaches. All of these issues impede the development of GNN-based solvers for SAT solving.

To systematically quantify the progress in this field and facilitate rapid, reproducible, and generalizable research, we propose **G4SATBench**, the first comprehensive benchmark study for SAT solving with GNNs. G4SATBench is characterized as follows:

- First, we construct a large and diverse collection of SAT datasets that includes instances from distinct sources and difficulty levels. Specifically, our benchmark consists of 7 different datasets from 3 benchmark families, including random instances, pseudo-industrial instances, and combinatorial problems. It not only covers a wide range of prior datasets but also introduces 3 levels of difficulty for each dataset to enable fine-grained analyses.

- Second, we re-implement various GNN-based SAT solvers with unified interfaces and configuration settings, establishing a general evaluation protocol for fair and comprehensive comparisons. Our framework allows for evaluating different GNN models in SAT solving with various prediction tasks, training objectives, and inference algorithms, encompassing the diverse learning frameworks employed in the existing literature.

- Third, we present baseline results and conduct thorough analyses of GNN-based SAT solvers, providing a detailed reference of prior work and laying a solid foundation for future research. Our evaluations assess the performances of different choices of GNN models (e.g., graph constructions, message-passing schemes) with particular attention to some critical parameters (e.g., message-passing iterations), as well as their generalization ability across different distributions.

- Lastly, we conduct a series of in-depth experiments to explore the learning abilities of GNN-based SAT solvers. Specifically, we compare the training and solving processes of GNNs with the heuristics employed in both CDCL and LS-based SAT solvers. Our experimental results reveal that *GNNs tend to develop a solving heuristic similar to greedy local search to find a satisfying assignment but fail to effectively learn the backtracking heuristic in the latent space.*

We believe that G4SATBench will enable the research community to make significant strides in understanding the capabilities and limitations of GNNs for solving SAT and facilitate further development in this area. Our codebase is available at `https://github.com/zhaoyu-li/G4SATBench`.

## 2    Related Work

**SAT solving with GNNs.**    Existing GNN-based SAT solvers can be broadly categorized into two branches [16]: *standalone neural solvers* and *neural-guided solvers*. Standalone neural solvers utilize GNNs to solve SAT instances directly. For example, a stream of research [6, 34, 21, 7, 35] focuses on predicting the satisfiability of a given formula, while several alternative approaches [1, 2, 30, 26, 42] aim to construct a satisfying assignment. Neural-guided solvers, on the other hand, integrate GNNs with modern SAT solvers, trying to improve their search heuristics with the prediction of GNNs. These methods typically train GNN models using supervised learning on some tasks such as unsat-core variable prediction [33, 38], satisfying assignment prediction [44], glue variable prediction [17], and assignment marginal prediction [28], or through reinforcement learning [43, 24] by modeling the entire search procedure as a Markov decision process. Despite the rich literature on SAT solving with GNNs, there is no benchmark study to evaluate and compare the performance of these GNN models. We hope the proposed G4SATBench would address this gap.

**SAT datasets.**    Several established SAT benchmarks, including the prestigious SATLIB [20] and the SAT Competitions over the years, have provided a variety of practical instances to assess the performance of modern SAT solvers. Regrettably, these datasets are not particularly amenable for GNNs to learn from, given their relatively modest scale (less than 100 instances for a specific domain) or overly extensive instances (exceeding 10 million variables and clauses). To address this issue,

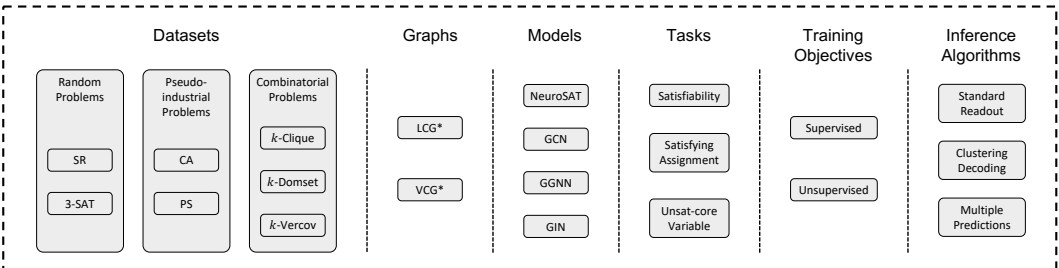

Figure 1: Framework overview of G4SATBench.

researchers have turned to synthetic SAT instance generators [34, 25, 14, 37], which allow for the creation of a flexible number of instances with customizable settings. However, most of the existing datasets generated from these sources are limited to a few domains (less than 3 generators), small in size (less than 10k instances), or easy in difficulty (less than 40 variables within an instance), and there is no standardized dataset for evaluation. In G4SATBench, we include a variety of synthetic generators with carefully selected configurations, aiming to construct a broad collection of SAT datasets that are highly conducive for training and evaluating GNNs.

## 3 Preliminaries

**The SAT problem.** In propositional logic, a Boolean formula is constructed from Boolean variables and logical operators such as conjunctions ($\wedge$), disjunctions ($\vee$), and negations ($\neg$). It is typical to represent Boolean formulas in conjunctive normal form (CNF), expressed as a conjunction of clauses, where each clause is a disjunction of literals, which can be either a variable or its negation. Given a CNF formula, the SAT problem is to determine if there exists an assignment of boolean values to its variables such that the formula evaluates to true. If this is the case, the formula is called satisfiable; otherwise, it is unsatisfiable. For a satisfiable instance, one is expected to construct a satisfying assignment to prove its satisfiability. On the other hand, for an unsatisfiable formula, one can find a minimal subset of clauses whose conjunction is still unsatisfiable. Such a set of clauses is termed the unsat core, and variables in the unsat core are referred to as unsat-core variables.

**Graph representations of CNF formulas.** Traditionally, a CNF formula can be represented using 4 types of graphs [4]: Literal-Clause Graph (LCG), Variable-Clause Graph (VCG), Literal-Incidence Graph (LIG), and Variable-Incidence Graph (VIG). The LCG is a bipartite graph with literal and clause nodes connected by edges indicating the presence of a literal in a clause. The VCG is formed by merging the positive and negative literals of the same variables in LCG. The LIG, on the other hand, only consists of literal nodes, with edges indicating co-occurrence in a clause. Lastly, the VIG is derived from LIG using the same merging operation as VCG.

## 4 G4SATBench: A Comprehensive Benchmark on GNNs for SAT Solving

The goal of G4SATBench is to establish a general framework that enables comprehensive comparisons and evaluations of various GNN-based SAT solvers. In this section, we will delve into the details of G4SATBench, including its datasets, GNN models, prediction tasks, as well as training and testing methodologies. The overview of the G4SATBench framework is shown in Figure 1.

### 4.1 Datasets

G4SATBench is built on a diverse set of synthetic CNF generators. It currently consists of 7 datasets sourced from 3 distinct domain areas: random problems, pseudo-industrial problems, and combinatorial problems. Specifically, we utilize the SR generator in NeuroSAT [34] and the 3-SAT generator in CNFGen [25] to produce random CNF formulas. For pseudo-industrial problems, we employ the Community Attachment (CA) model [14] and the Popularity-Similarity (PS) model [15],

which generate synthetic instances that exhibit similar statistical features, such as the community and the locality, to those observed in real-world industrial SAT instances. For combinatorics, we resort to 3 synthetic generators in CNFGen [25] to create SAT instances derived from the translation of $k$-Clique, $k$-Dominating Set, and $k$-Vertex Cover problems.

In addition to the diversity of datasets, G4SATBench offers distinct difficulty levels for all datasets to enable fine-grained analyses. These levels include easy, medium, and hard, with the latter representing more complex problems with increased instance sizes. For example, the easy SR dataset contains instances with 10 to 40 variables, the medium SR dataset contains formulas with 40 to 200 variables, and the hard SR dataset consists of formulas with variables ranging from 200 to 400. For each easy and medium dataset, we generate 80k pairs of satisfiable and unsatisfiable instances for training, 10k pairs for validation, and 10k pairs for testing. For each hard dataset, we produce 10k testing pairs. It is also worth noting that the parameters for our synthetic generators are meticulously selected to avoid generating trivial cases. For instance, we produce random 3-SAT formulas at the phase-transition region where the relationship between the number of clauses $(m)$ and variables $(n)$ is $m = 4.258n + 58.26n^{-2/3}$ [10], and utilize the $v$ vertex Erdős-Rényi graph with an edge probability of $p = \binom{v}{k}^{-1/\binom{v}{2}}$ to generate $k$-Clique problems, making the expected number of $k$-Cliques in a graph equals 1 [5]. To provide a detailed characterization of our generated datasets, we compute several statistics of the SAT instances across difficulty levels in G4SATBench. For more information about the generators we used and the dataset statistics, please refer to Appendix A.

## 4.2 GNN Baselines

**Graph constructions.** It is important to note that traditional graph representations of a CNF formula often lack the requisite details for optimally constructing GNNs. Specifically, the LIG and VIG exclude clause-specific information, while the LCG and VIG fail to differentiate between positive and negative literals of the same variable. To address these limitations, existing approaches typically build GNN

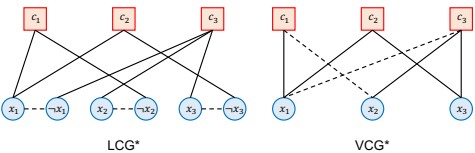

Figure 2: LCG* and VCG* of the CNF formula $(x_1 \vee \neg x_2) \wedge (x_1 \vee x_3) \wedge (\neg x_1 \vee x_2 \vee x_3)$.

models on the refined versions of the LCG and VCG encodings. In the LCG, a new type of edge is added between each literal and its negation, while the VCG is modified by using two types of edges to indicate the polarities of variables within a clause. These modified encodings are termed the LCG* and VCG* respectively, and an example of them is shown in Figure 2.

**Message-passing schemes.** G4SATBench enables performing various *heterogeneous* message-passage algorithms between neighboring nodes on the LCG* or VCG* encodings of a CNF formula. For the sake of illustration, we will take GNN models on the LCG* as an example. We first define a $d$-dimensional embedding for every literal node and clause node, denoted by $h_l$ and $h_c$ respectively. Initially, all these embeddings are assigned to two learnable vectors $h_l^0$ and $h_c^0$, depending on their node types. At the $k$-th iteration of message passing, these hidden representations are updated as:

$$h_c^{(k)} = \text{UPD}\left(\underset{l \in \mathcal{N}(c)}{\text{AGG}}\left(\left\{\text{MLP}_l\left(h_l^{(k-1)}\right)\right\}\right), h_c^{(k-1)}\right),$$

$$h_l^{(k)} = \text{UPD}\left(\underset{c \in \mathcal{N}(l)}{\text{AGG}}\left(\left\{\text{MLP}_c\left(h_c^{(k-1)}\right)\right\}\right), h_{\neg l}^{(k-1)}, h_l^{(k-1)}\right),$$

where $\mathcal{N}(\cdot)$ denotes the set of neighbor nodes, $\text{MLP}_l$ and $\text{MLP}_c$ are two different multi-layer perceptions (MLPs), $\text{UPD}(\cdot)$ is the update function, and $\text{AGG}(\cdot)$ is the aggregation function. Most GNN models on LCG* use Equation 1 with different choices of the update function and aggregation function. For instance, NeuroSAT employs LayerNormLSTM [3] as the update function and summation as the aggregation function. In G4SATBench, we provide a diverse range of GNN models, including NeuroSAT [34], Graph Convolutional Network (GCN) [23], Gated Graph Neural Network (GGNN) [27], and Graph Isomorphism Network (GIN) [41], on the both LCG* and VCG*. More details of these GNN models are included in Appendix B.

### 4.3 Supported Tasks, Training and Testing Settings

**Prediction tasks.** In G4SATBench, we support three essential prediction tasks for SAT solving: satisfiability prediction, satisfying assignment prediction, and unsat-core variable prediction. These tasks are widely used in both standalone neural solvers and neural-guided solvers. Technically, we model satisfiability prediction as a binary graph classification task, where 1/0 denotes the satisfiability/unsatisfiability of the given SAT instance $\phi$. Here, we take GNN models on the LCG* as an example. After $T$ iterations of message passing, we obtain the graph embedding by applying mean pooling on all literal embeddings, and then predict the satisfiability using an MLP followed by the sigmoid function $\sigma$:

$$y_\phi = \sigma\left(\text{MLP}\left(\text{MEAN}\left(\{h_l^{(T)}, l \in \phi\}\right)\right)\right). \tag{2}$$

For satisfying assignment prediction and unsat-core variable prediction, we formulate them as binary node classification tasks, predicting the label for each variable in the given CNF formula $\phi$. In the case of GNNs on the LCG*, we concatenate the embeddings of each pair of literals $h_l$ and $h_{\neg l}$ to construct the variable embedding, and then readout using an MLP and the sigmoid function $\sigma$:

$$y_v = \sigma\left(\text{MLP}\left(\left[h_l^{(T)}, h_{\neg l}^{(T)}\right]\right)\right). \tag{3}$$

**Training objectives.** To train GNN models on the aforementioned tasks, one common approach is to minimize the binary cross-entropy loss between the predictions and the ground truth labels. In addition to supervised learning, G4SATBench supports two unsupervised training paradigms for satisfying assignment prediction [1, 30]. The first approach aims to differentiate and maximize the satisfiability value of a CNF formula [1]. It replaces the $\neg$ operator with the function $N(a) = 1 - a$ and uses smooth max and min functions to replace the $\vee$ and $\wedge$ operators. The smooth max and min functions are defined as follows:

$$S_{max}(x_1, x_2, \ldots, x_d) = \frac{\sum_{i=1}^d x_i \cdot e^{x_i/\tau}}{\sum_{i=1}^d e^{x_i/\tau}}, \quad S_{min}(x_1, x_2, \ldots, x_d) = \frac{\sum_{i=1}^d x_i \cdot e^{-x_i/\tau}}{\sum_{i=1}^d e^{-x_i/\tau}}, \tag{4}$$

where $\tau \geq 0$ is the temperature parameter. Given a predicted soft assignment $x = (x_1, x_2, \ldots, x_n)$, we evaluate its satisfiability value $S(x)$ using the smoothed version of logical operators and minimize the following loss function:

$$\mathcal{L}_\phi(x) = \frac{(1 - S(x))^\kappa}{(1 - S(x))^\kappa + S(x)^\kappa}. \quad (\kappa \geq 1 \text{ is a predefined constant}) \tag{5}$$

The second unsupervised loss is defined as follows [30]:

$$V_c(x) = 1 - \prod_{i \in c^+}(1 - x_i) \prod_{i \in c^-} x_i, \quad \mathcal{L}_\phi(x) = -\log\left(\prod_{c \in \phi} V_c(x)\right) = -\sum_{c \in \phi} \log\left(V_c(x)\right), \tag{6}$$

where $c^+$ and $c^-$ are the sets of variables that occur in the clause $c$ in positive and negative form respectively. Note that these two losses reach the minimum only when the prediction $x$ is a satisfying assignment, thus minimizing such losses could help to construct a possible satisfying assignment.

**Inference algorithms.** In addition to using the standard readout process like training, G4SATBench offers two alternative inference algorithms for satisfying assignment prediction [34, 2]. The first method performs 2-clustering on the literal embeddings to obtain two centers $\Delta_1$ and $\Delta_2$ and then partitions the positive and negative literals of each variable into distinct groups based on the predicate $||x_i - \Delta_1||^2 + ||\neg x_i - \Delta_2||^2 < ||x_i - \Delta_2||^2 + ||\neg x_i - \Delta_1||^2$ [34]. This allows the construction of two possible assignments by mapping one group of literals to true. The second approach is to employ the readout function at each iteration of message passing, resulting in multiple assignment predictions for a given instance [2].

**Evaluation metrics.** For satisfiability prediction and unsat-core variable prediction, we report the classification accuracy of each GNN model in G4SATBench. For satisfying assignment prediction, we report the solving accuracy of the predicted assignments. If multiple assignments are predicted for a SAT instance, the instance is considered solved if any of the predictions satisfy the formula.

## 5 Benchmarking Evaluation on G4SATBench

In this section, we present the benchmarking results of G4SATBench. To ensure a fair comparison, we conduct a grid search to tune the hyperparameters of each GNN baseline. The best checkpoint for each GNN model is selected based on its performance on the validation set. To mitigate the impact of randomness, we use 3 different random seeds to repeat the experiment in each setting and report the average performance. Each experiment is performed on a single RTX8000 GPU and 16 AMD EPYC 7502 CPU cores, and the total time cost is approximately 8,000 GPU hours. For detailed experimental setup and hyperparameters, please refer to Appendix C.1.

### 5.1 Satisfiability Prediction

**Evaluation on the same distribution.**  Table 1 shows the benchmarking results of each GNN baseline when trained and evaluated on datasets possessing identical distributions. All GNN models exhibit strong performance across most easy and medium datasets, except for the medium SR dataset. This difficulty can be attributed to the inherent characteristic of this dataset, which includes satisfiable and unsatisfiable pairs of medium-sized instances distinguished by just a single differing literal. Such a subtle difference presents a substantial challenge for GNN models in satisfiability classification. Among all GNN models, the different graph constructions do not seem to have a significant impact on the results, and NeuroSAT (on LCG*) and GGNN (on VCG*) achieve the best overall performance.

Table 1: Results on the datasets of the same distribution.

| Graph | Method | Easy Datasets | | | | | | | Medium Datasets | | | | | | |
|---|---|---|---|---|---|---|---|---|---|---|---|---|---|---|---|
| | | SR | 3-SAT | CA | PS | $k$-Clique | $k$-Domset | $k$-Vercov | SR | 3-SAT | CA | PS | $k$-Clique | $k$-Domset | $k$-Vercov |
| LCG* | NeuroSAT | 96.00 | **96.33** | **98.83** | 96.59 | 97.92 | **99.77** | **99.99** | **78.02** | **84.90** | 99.57 | 96.81 | 89.39 | 99.67 | 99.80 |
| | GCN | 94.43 | 94.47 | 98.79 | **97.53** | 98.24 | 99.59 | 99.98 | 69.39 | 82.67 | 99.53 | 96.16 | 85.72 | 99.16 | 99.74 |
| | GGNN | **96.36** | 95.70 | 98.81 | 97.47 | **98.80** | 99.77 | 99.97 | 71.44 | 83.45 | 99.50 | 96.21 | 81.20 | **99.69** | **99.83** |
| | GIN | 95.78 | 95.37 | 98.14 | 96.98 | 97.60 | 99.71 | 99.97 | 70.54 | 82.80 | 99.49 | 95.80 | 83.87 | 99.61 | 99.62 |
| VCG* | GCN | 93.19 | 94.92 | 97.82 | 95.79 | 98.72 | 99.54 | **99.99** | 66.35 | 83.75 | 99.49 | 95.48 | 82.99 | 99.42 | **99.89** |
| | GGNN | **96.75** | **96.25** | **98.77** | 96.44 | **98.88** | **99.68** | 99.98 | **77.12** | 85.11 | **99.57** | 96.48 | 83.63 | **99.62** | 98.92 |
| | GIN | 96.04 | 95.71 | 98.47 | **96.95** | 97.33 | 99.59 | 99.98 | 73.56 | **85.26** | 99.49 | **96.55** | 89.41 | 99.38 | 99.80 |

**Evaluation across different distributions.**  To assess the generalization ability of GNN models, we evaluate the performance of NeuroSAT (on LCG*) and GGNN (on VCG*) across different datasets and difficulty levels. As shown in Figure 3 and Figure 4, NeuroSAT and GGNN struggle to generalize effectively to datasets distinct from their training data in most cases. However, when trained on the SR dataset, they exhibit better generalization performance across different datasets. Furthermore, while both GNN models demonstrate limited generalization to larger formulas beyond their training data, they perform relatively better on smaller instances. These observations suggest that the generalization performance of GNN models for satisfiability prediction is influenced by the distinct nature and complexity of its training data. Training on more challenging instances could potentially enhance their generalization ability.

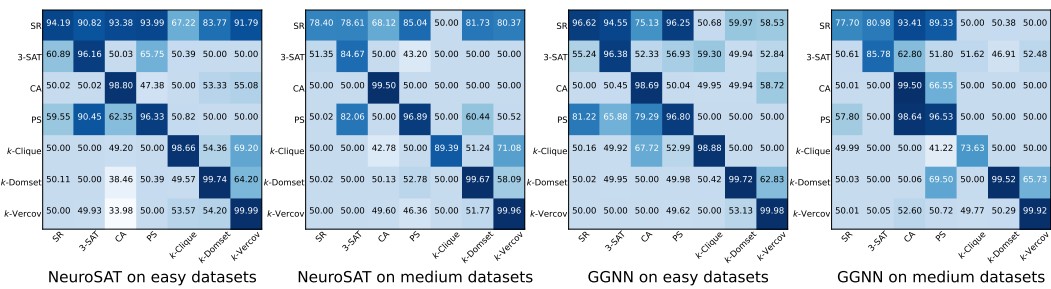

Figure 3: Results across different datasets. The x-axis denotes testing datasets and the y-axis denotes training datasets.

Due to the limited space, Figure 4 exclusively displays the performance of NeuroSAT and GGNN on the SR and 3-SAT datasets. Comprehensive results on the other five datasets, as well as the experimental results on different massage passing iterations, are provided in Appendix C.2.

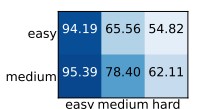
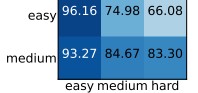
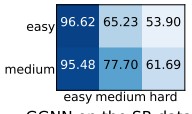
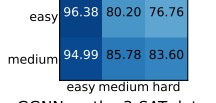

| | | | | | | |
|---|---|---|---|---|---|---|
| NeuroSAT on the SR dataset | | NeuroSAT on the 3-SAT dataset | | GGNN on the SR dataset | | GGNN on the 3-SAT dataset |

Figure 4: Results across different difficulty levels. The x-axis denotes testing datasets and the y-axis denotes training datasets.

## 5.2 Satisfying Assignment Prediction

**Evaluation with different training losses.** Table 2 presents the benchmarking results of each GNN baseline across three different training objectives. Interestingly, the unsupervised training methods outperform the supervised learning approach across the majority of datasets. We hypothesize that this is due to the presence of multiple satisfying assignments in most satisfiable instances. Supervised training tends to bias GNN models towards learning a specific satisfying solution, thereby neglecting the exploration of other feasible ones. This bias may compromise the models' ability to generalize effectively. Such limitations become increasingly apparent when the space of satisfying solutions is much larger, as seen in the medium CA and PS datasets. Additionally, it is noteworthy that employing $UNS_1$ as the loss function can result in instability during the training of some GNN models, leading to a failure to converge in some cases. Conversely, using $UNS_2$ loss demonstrates strong and stable performance across all datasets.

Table 2: Results on the datasets of the same distribution with different training losses. The top and bottom 7 rows represent the results for easy and medium datasets, respectively. SUP denotes the supervised loss, $UNS_1$ and $UNS_2$ correspond to the unsupervised losses defined in Equation 5 and Equation 6, respectively. The symbol "-" indicates that some seeds failed during training. Note that only satisfiable instances are evaluated in this experiment.

| Graph | Method | SR | | | 3-SAT | | | CA | | | PS | | | $k$-Clique | | | $k$-Domset | | | $k$-Vercov | | |
|---|---|---|---|---|---|---|---|---|---|---|---|---|---|---|---|---|---|---|---|---|---|---|
| | | SUP | $UNS_1$ | $UNS_2$ | SUP | $UNS_1$ | $UNS_2$ | SUP | $UNS_1$ | $UNS_2$ | SUP | $UNS_1$ | $UNS_2$ | SUP | $UNS_1$ | $UNS_2$ | SUP | $UNS_1$ | $UNS_2$ | SUP | $UNS_1$ | $UNS_2$ |
| LCG* | NeuroSAT | **88.47** | 82.30 | 79.79 | 78.39 | 80.23 | 80.59 | 0.27 | 82.17 | **89.34** | 39.18 | **89.23** | 88.79 | 66.30 | **88.34** | 63.43 | 69.61 | 96.74 | **98.85** | 85.15 | 99.36 | **99.73** |
| | GCN | 83.74 | 73.09 | 77.02 | 70.34 | 74.79 | 75.31 | 0.17 | 75.30 | 82.41 | 39.66 | 82.75 | 84.89 | 63.85 | 82.60 | 86.17 | 59.29 | 97.50 | 97.55 | 76.83 | 99.16 | 99.28 |
| | GGNN | 84.13 | 76.39 | 78.75 | 72.87 | 76.55 | 76.42 | 0.29 | 78.13 | 84.08 | 38.82 | 84.44 | 86.29 | 60.80 | 84.60 | 87.12 | 68.36 | 97.49 | 98.06 | 82.06 | - | 99.34 |
| | GIN | 83.81 | 81.45 | 80.39 | 73.99 | 78.47 | 76.24 | 0.20 | 78.44 | 85.15 | 39.13 | 85.31 | 85.43 | 56.85 | 84.48 | 85.11 | 68.93 | 96.99 | 97.43 | 81.49 | 99.28 | 99.38 |
| VCG* | GCN | 83.38 | 84.19 | 78.00 | 76.60 | 84.42 | 79.23 | 14.98 | 76.64 | 83.79 | 51.48 | 85.88 | 83.06 | 56.27 | 85.28 | 86.91 | 66.32 | 97.62 | 96.74 | 78.67 | - | 93.51 |
| | GGNN | 86.30 | 87.16 | 81.00 | 77.96 | **88.97** | 79.32 | 15.11 | 76.32 | 83.12 | 47.67 | 86.85 | 87.17 | 66.86 | 86.31 | 87.48 | 66.42 | - | 98.42 | 82.61 | - | 99.52 |
| | GIN | 84.61 | 89.56 | 83.27 | 79.23 | 87.65 | 81.72 | 17.81 | 83.28 | 86.03 | 48.92 | 91.21 | 85.65 | 66.07 | 86.12 | 88.09 | 67.67 | - | - | 81.01 | 99.38 | 99.41 |
| LCG* | NeuroSAT | **34.97** | 25.00 | 37.25 | 20.07 | 30.40 | **41.61** | 0.00 | 35.45 | **70.83** | 3.64 | 60.28 | **71.03** | 56.61 | 41.45 | 32.48 | 52.09 | 95.06 | **96.18** | 74.77 | 67.44 | 95.99 |
| | GCN | 13.19 | 13.76 | 19.21 | 8.87 | 20.50 | 24.58 | 0.00 | 30.20 | 54.04 | 1.45 | 45.16 | 56.29 | 55.36 | 61.82 | 66.33 | 43.50 | 92.86 | 94.89 | 67.83 | - | 93.84 |
| | GGNN | 14.15 | 16.55 | 21.18 | 7.96 | 22.84 | 25.68 | 0.00 | 28.12 | 50.66 | 2.33 | 44.89 | 57.96 | 52.35 | 54.29 | **68.91** | 49.07 | - | 92.26 | 69.21 | 66.37 | 94.30 |
| | GIN | 15.36 | 18.60 | 22.17 | 9.66 | 21.38 | 24.93 | 0.00 | 35.76 | 57.81 | 2.02 | 43.43 | 57.62 | 53.07 | 44.60 | 66.32 | 44.39 | 93.3 | 93.82 | 70.59 | 55.59 | 95.69 |
| VCG* | GCN | 20.59 | 9.21 | 22.44 | 12.48 | 17.00 | 29.53 | 0.44 | 39.04 | 48.99 | 2.29 | 35.99 | 55.46 | 46.09 | 25.90 | 68.62 | 46.96 | - | 96.46 | 69.15 | - | 96.46 |
| | GGNN | 28.04 | 27.72 | 33.37 | 16.46 | 29.65 | 35.95 | 0.56 | 48.13 | 49.93 | 3.12 | 51.73 | 65.11 | 44.26 | 48.92 | 56.43 | 51.01 | - | - | 71.97 | - | 95.23 |
| | GIN | 26.73 | 26.48 | 31.97 | 14.64 | 26.86 | 35.81 | 0.64 | 44.06 | 63.84 | 3.38 | 58.03 | 64.66 | 55.47 | 56.97 | 67.78 | 46.98 | - | 95.28 | 69.40 | - | **96.96** |

In addition to evaluating the performance of GNN models under various training loss functions, we extend our analysis to explore how these models perform across different data distributions and under various inference algorithms. Furthermore, we assess the robustness of these GNN models when trained on noisy datasets that include unsatisfiable instances in an unsupervised fashion. For detailed results of these evaluations, please refer to Appendix C.3.

## 5.3 Unsat-core Variable Prediction

**Evaluation on the same distribution.** The benchmarking results presented in Table 3 exhibit the superior performance of all GNN models on both easy and medium datasets, with NeuroSAT consistently achieving the best results across most datasets. It is important to note that the primary objective of predicting unsat-core variables is not to solve SAT problems directly but to provide valuable guidance for enhancing the backtracking search process. As such, even imperfect predictions - for instance, those with a classification accuracy of 90% - have been demonstrated to be sufficiently effective in improving the search heuristics employed by modern CDCL-based SAT solvers, as indicated by previous studies [33, 38].

We also conduct experiments to evaluate the generalization ability of GNN models on unsat-core variable prediction. Please see appendix C.4 for details.

Table 3: Results on the datasets of the same distribution. Only unsatisfiable instances are evaluated.

| Graph | Method | Easy Datasets | | | | | | | Medium Datasets | | | | | | |
|---|---|---|---|---|---|---|---|---|---|---|---|---|---|---|---|
| | | SR | 3-SAT | CA | PS | $k$-Clique | $k$-Domset | $k$-Vercov | SR | 3-SAT | CA | PS | $k$-Clique | $k$-Domset | $k$-Vercov |
| LCG* | NeuroSAT | **90.76** | **94.43** | **83.69** | **86.20** | **99.93** | 95.80 | **94.47** | **90.07** | 99.65 | 85.73 | 88.53 | **99.97** | **97.90** | **99.10** |
| | GCN | 89.17 | 94.35 | 82.89 | 85.32 | 99.93 | 95.74 | 94.43 | 88.11 | 99.65 | 85.71 | 87.70 | 99.96 | 97.89 | 99.10 |
| | GGNN | 90.02 | 94.38 | 83.59 | 86.03 | 99.93 | 95.79 | 94.46 | 89.05 | 99.65 | 85.69 | 87.95 | 99.96 | 97.89 | 99.09 |
| | GIN | 89.29 | 94.33 | 83.71 | 85.97 | 99.93 | **95.81** | 94.47 | 88.85 | 99.65 | 85.71 | 87.92 | 99.96 | 97.89 | 99.09 |
| VCG* | GCN | 88.57 | 94.34 | 83.17 | 85.27 | 99.93 | **95.79** | 94.46 | 88.17 | 99.65 | 85.70 | 87.37 | 99.96 | **97.90** | 99.09 |
| | GGNN | **89.57** | **94.37** | **83.50** | **85.84** | **99.93** | 95.81 | **94.49** | 88.84 | 99.65 | 85.68 | 88.03 | **99.98** | 97.90 | **99.10** |
| | GIN | 89.50 | 94.35 | 83.23 | 85.69 | 99.93 | 95.79 | 94.47 | **89.51** | 99.65 | **85.72** | **88.13** | 99.96 | 97.89 | 99.10 |

# 6 Advancing Evaluation on G4SATBench

To gain deeper insights into how GNNs tackle the SAT problem, we conduct comprehensive comparative analyses between GNN-based SAT solvers and the CDCL and LS heuristics in this section. Since these search heuristics aim to solve a SAT instance directly, our focus only lies on the tasks of (**T1**) satisfiability prediction and (**T2**) satisfying assignment prediction (with $UNS_2$ as the training loss). We employ NeuroSAT (on LCG*) and GGNN (on VCG*) as our GNN models and experiment on the SR and 3-SAT datasets. Detailed experimental settings are included in Appendix D.

## 6.1 Comparison with the CDCL Heuristic

**Evaluation on the clause-learning augmented instances.** CDCL-based SAT solvers enhance backtracking search with conflict analysis and clause learning, enabling efficient exploration of the search space by iteratively adding "learned clauses" to avoid similar conflicts in future searches [36]. To assess whether GNN-based SAT solvers can learn and benefit from the backtracking search (with CDCL) heuristic, we augment the original formulas in the datasets with learned clauses and evaluate GNN models on these clause-learning augmented instances.

Table 4 shows the testing results on augmented SAT datasets. Notably, training on the augmented instances leads to significant improvements in both satisfiability prediction and satisfying assignment prediction. These improvements can be attributed to the presence of "learned clauses" that effectively modify the graph structure of the original formulas, thereby facilitating GNNs to solve them with relative ease. However, despite the augmented instances being easily solvable using the backtracking search within a few search steps, GNN models fail to effectively handle these instances when trained on the original instances. These findings suggest that GNNs may not explicitly learn the backtracking search heuristic when trained for satisfiability prediction or satisfying assignment prediction.

Table 4: Results on augmented datasets. Values inside/outside parentheses denote the results of models trained on augmented/original instances.

| Task | Method | Easy Datasets | | Medium Datasets | |
|---|---|---|---|---|---|
| | | SR | 3-SAT | SR | 3-SAT |
| T1 | NeuroSAT | 100.00 (96.78) | 100.00 (96.06) | 100.00 (84.57) | 96.78 (84.85) |
| | GGNN | 100.00 (97.66) | 100.00 (95.46) | 100.00 (84.01) | 96.29 (85.80) |
| T2 | NeuroSAT | 85.05 (83.28) | 83.50 (81.04) | 51.95 (45.51) | 39.00 (16.52) |
| | GGNN | 85.35 (83.42) | 81.56 (79.99) | 44.18 (40.09) | 34.67 (14.75) |

Table 5: Results using contrastive pretraining. Values in parentheses denote the difference between the results without pretraining.

| Task | Method | Easy Datasets | | Medium Datasets | |
|---|---|---|---|---|---|
| | | SR | 3-SAT | SR | 3-SAT |
| T1 | NeuroSAT | 96.68 (+0.68) | 96.23 (-0.10) | 78.31 (+0.29) | 85.02 (+0.12) |
| | GGNN | 96.46 (-0.29) | 96.45 (+0.20) | 76.34 (-0.78) | 85.17 (+0.06) |
| T2 | NeuroSAT | 80.54 (+0.75) | 79.71 (-0.88) | 36.42 (-0.83) | 41.23 (-0.38) |
| | GGNN | 80.66 (-0.34) | 79.23 (-0.09) | 33.44 (+0.07) | 36.39 (+0.44) |

**Evaluation with contrastive pretraining.** Observing that GNN models exhibit superior performance on clause-learning augmented SAT instances, there is potential to improve the performance of GNNs by learning a latent representation of the original formula similar to its augmented counterpart. Motivated by this, we also experiment with a contrastive learning approach (i.e., SimCLR [8]) to pretrain the representation of CNF formulas to be close to their augmented ones [11], trying to embed the CDCL heuristic in the latent space through representation learning.

The results of contrastive pretraining are presented in Table 5. In contrast to the findings in [11], our results show limited performance improvement through contrastive pretraining, indicating that GNN models still encounter difficulties in effectively learning the CDCL heuristic in the latent space. This observation aligns with the conclusions drawn in [9], which highlight that static GNNs may fail

to exactly replicate the same search operations due to the dynamic changes in the graph structure introduced by the clause learning technique.

## 6.2 Comparison with the LS Heuristic

**Evaluation with random initialization.** LS-based SAT solvers typically begin by randomly initializing an assignment and then iteratively flip variables guided by specific heuristics until reaching a satisfying assignment. To compare the behaviors of GNNs with this solving procedure, we first conduct an evaluation of GNN models with randomized initial embeddings in both training and testing, emulating the initialization of LS SAT solvers.

The results presented in Table 6 demonstrate that using random initialization has a limited impact on the overall performances of GNN-based SAT solvers. This suggests that GNN models do not aim to learn a fixed latent representation for each formula in SAT solving. Instead, they have developed a solving strategy that effectively exploits the inherent graph structure of each SAT instance.

Table 6: Results using random initialization. Values in parentheses denote the difference between the results with learned initialization.

| Task | Method | Easy Datasets | | Medium Datasets | |
|------|--------|------|------|------|------|
| | | SR | 3-SAT | SR | 3-SAT |
| T1 | NeuroSAT | 97.24 (+1.24) | 96.44 (+0.11) | 77.29 (-0.91) | 84.85 (-0.05) |
| | GGNN | 96.78 (+0.03) | 96.38 (+0.13) | 76.97 (-0.15) | 85.80 (+0.69) |
| T2 | NeuroSAT | 79.09 (-0.70) | 80.79 (+0.20) | 37.27 (+0.02) | 40.75 (-0.86) |
| | GGNN | 80.10 (-0.90) | 79.83 (+0.51) | 32.85 (-0.52) | 36.59 (+0.64) |

**Evaluation on the predicted assignments.** Under random initialization, we further analyze the solving strategies of GNNs by evaluating their predicted assignments decoded from the latent space. For the task of satisfiability prediction, we employ the 2-clustering decoding algorithm to extract the predicted assignments from the literal embeddings of NeuroSAT at each iteration of message passing. For satisfying assignment prediction, we evaluate both NeuroSAT and GGNN using multiple-prediction decoding. Our evaluation focuses on three key aspects: (a) the number of distinct predicted assignments, (b) the number of flipped variables between two consecutive iterations, and (c) the number of unsatisfiable clauses associated with the predicted assignments.

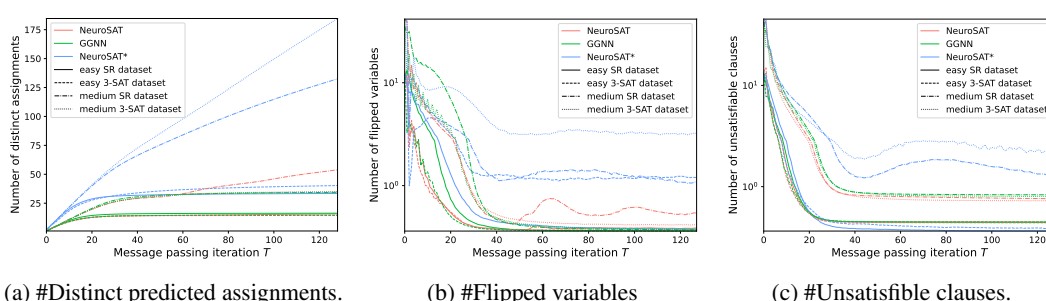

(a) #Distinct predicted assignments.    (b) #Flipped variables    (c) #Unsatisfible clauses.

Figure 5: Results on the predicted assignments with the increased message passing iteration $T$. NeuroSAT* refers to the model trained for satisfiability prediction.

As shown in Figure 5, all three GNN models initially generate a wide array of assignment predictions by flipping a considerable number of variables, resulting in a notable reduction in the number of unsatisfiable clauses. However, as the iterations progress, the number of flipped variables diminishes substantially, and most GNN models eventually converge towards predicting a specific assignment or making minimal changes to their predictions when there are no or very few unsatisfiable clauses remaining. This trend is reminiscent of the greedy solving strategy adopted by the LS solver GSAT [32], where changes are made to minimize the number of unsatisfied clauses in the new assignment. However, unlike GSAT's approach of flipping one variable at a time and incorporating random selection to break ties, GNN models simultaneously modify multiple variables and potentially converge to a particular unsatisfied assignment and find it challenging to deviate from such a prediction. It is also noteworthy that despite being trained for satisfiability prediction, NeuroSAT* demonstrates similar behavior to the GNN models trained for assignment prediction. This observation indicates that GNNs also learn to search for a satisfying assignment implicitly in the latent space while performing satisfiability prediction.

## 7 Discussions

**Limitations and future work.** While G4SATBench represents a significant step in evaluating GNNs for SAT solving, there are still some limitations and potential future directions to consider. Firstly, G4SATBench primarily focuses on evaluating standalone neural SAT solvers, excluding the exploration of neural-guided SAT solvers that integrate GNNs with search-based SAT solvers. It also should be emphasized that the instances included in G4SATBench are relatively small compared to most practical instances found in real-world applications, where GNN models alone are not sufficient for solving such large-scale instances. Future research could explore techniques to effectively leverage GNNs in combination with modern SAT solvers to scale up to real-world instances. Secondly, G4SATBench benchmarks general GNN models on the LCG* and VCG* graph representations for SAT solving, but does not consider sophisticated GNN models designed for specific graph constructions in certain domains, such as Circuit SAT problems. Investigating domain-specific GNN models tailored to the characteristics of specific problems could lead to improved performance in specialized instances. Lastly, all existing GNN-based SAT solvers in the literature are static GNNs, which have limited learning ability to capture the CDCL heuristic. Exploring dynamic GNN models that can effectively learn the CDCL heuristic is also a potential direction for future research.

**Conclusion.** In this work, we present G4SATBench, a groundbreaking benchmark study that comprehensively evaluates GNN models in SAT solving. G4SATBench offers curated synthetic SAT datasets sourced from various domains and difficulty levels and benchmarks a wide range of GNN-based SAT solvers under diverse settings. Our empirical analysis yields valuable insights into the performances of GNN-based SAT solvers and further provides a deeper understanding of their capabilities and limitations. We hope the proposed G4SATBench will serve as a solid foundation for GNN-based SAT solving and inspire future research in this exciting field.

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
