# A  Datasets

**Generators.** To generate high-quality SAT datasets that do not contain trivial instances, we have employed a rigorous process of selecting appropriate parameters for each CNF generator in G4SATBench. Table 7 provides detailed information about the generators we have used.

Table 7: Details of the synthetic generators employed in G4SATBench.

| Dataset | Description | Parameters | Notes |
|---|---|---|---|
| SR | The SR dataset is composed of pairs of satisfiable and unsatisfiable formulas, with the only difference between each pair being the polarity of a single literal. Given the number of variables $n$, the synthetic generator iteratively samples $k = 1 + \text{Bernoulli}(b) + \text{Geometric}(g)$ variables uniformly at random without replacement and negates each one with independent probability 50% to build a clause. This procedure continues until the generated formula is unsatisfiable. The satisfiable instance is then constructed by negating the first literal in the last clause of the unsatisfiable one. | General: $b = 0.3, g = 0.4$, Easy dataset: $n \sim \text{Uniform}(10, 40)$, Medium dataset: $n \sim \text{Uniform}(40, 200)$, Hard dataset: $n \sim \text{Uniform}(200, 400)$ | The sampling parameters are the same as the original paper [34]. |
| 3-SAT | The 3-SAT dataset comprises CNF formulas at the phase transition, where the proportion of generated satisfiable and unsatisfiable formulas is roughly equal. Given the number of variables $n$ and clauses $m$, the synthetic generator iteratively samples three variables (and their polarities) uniformly at random until $m$ clauses are obtained. | General: $m = 4.258n + 58.26n^{-2/3}$, Easy dataset: $n \sim \text{Uniform}(10, 40)$, Medium dataset: $n \sim \text{Uniform}(40, 200)$, Hard dataset: $n \sim \text{Uniform}(200, 300)$ | The parameter $m$ is the same as the paper [10] |
| CA | The CA dataset contains SAT instances that are designed to mimic the community structures and modularity features found in real-world industrial instances. Given variable number $n$, clause number $m$, clause size $k$, community number $c$, and modularity $Q$, the synthetic generator iteratively selects $k$ literals in the same community uniformly at random with probability $P = Q + 1/c$ and selects $k$ literals in the distinct community uniformly at random with probability $1 - P$ to build a clause and repeat for $m$ times to construct a CNF formula. | General: $m \sim \text{Uniform}(13n, 15n)$, $k \sim \text{Uniform}(4, 5)$, $c \sim \text{Uniform}(3, 10)$, $Q \sim \text{Uniform}(0.7, 0.9)$ Easy dataset: $n \sim \text{Uniform}(10, 40)$, Medium dataset: $n \sim \text{Uniform}(40, 200)$, Hard dataset: $n \sim \text{Uniform}(200, 400)$ | The parameters are selected based on the experiments in the original paper [14] and our own study to ensure that the generated SAT instances have a balance of satisfiability and unsatisfiability. |
| PS | PS dataset encompasses SAT instances with a power-law distribution in the number of variable occurrences (popularity), and good clustering between them (similarity). Given variable number $n$, clause number $m$, and average clause size $k$, the synthetic generator first assigns random angles $\theta_i, \theta_j \in [0, 2\pi]$ to each variable $i$ and each clause $j$, and then randomly samples variable $i$ in clause $j$ with the probability $P = 1/(1 + (i^\beta j^{\beta'} \theta_{ij}/R)^T)$. Here, $\theta_{ij} = \pi - |\pi - |\theta_i - \theta_j||$ is the angle between variable $i$ and clause $j$. The exponent parameters $\beta$ and $\beta'$ control the power-law distribution of variable occurrences and clause size respectively. The temperature parameter $T$ controls the sharpness of the probability distribution, while $R$ is an approximate normalization constant that ensures the average number of selected edges is $km$. | General: $m \sim \text{Uniform}(6n, 8n)$, $k \sim \text{Uniform}(4, 5)$, $\beta \sim \text{Uniform}(0, 1)$, $\beta' = 1$, $c \sim \text{Uniform}(3, 10)$, $T \sim \text{Uniform}(0.75, 1.5)$ Easy dataset: $n \sim \text{Uniform}(10, 40)$, Medium dataset: $n \sim \text{Uniform}(40, 200)$, Hard dataset: $n \sim \text{Uniform}(200, 300)$ | The parameters are selected based on the experiments in the original paper [15] and our own study to ensure that the generated SAT instances have a balance of satisfiability and unsatisfiability. |
| $k$-Clique | The $k$-Clique dataset includes SAT instances that encode the $k$-Clique problem, which involves determining whether there exists a clique (i.e., a subset of vertices that are all adjacent to each other) with $v$ vertices in a given graph. Given the number of cliques $k$, the synthetic generator produces an Erdős-Rényi graph with $v$ vertices and a given edge probability $p$ and then transforms the corresponding $k$-Clique problem into a SAT instance. | General: $p = \binom{v}{k}^{-1/\binom{v}{2}}$, Easy dataset: $v \sim \text{Uniform}(5, 15)$, $k \sim \text{Uniform}(3, 4)$, Medium dataset: $v \sim \text{Uniform}(15, 20)$, $k \sim \text{Uniform}(3, 5)$, Hard dataset: $v \sim \text{Uniform}(20, 25)$, $k \sim \text{Uniform}(4, 6)$ | The parameter $p$ is selected based on the paper [5], making the expected number of $k$-Cliques in the generated graph equals 1. |
| $k$-Domset | The $k$-Domset dataset contains SAT instances that encode the $k$-Dominating Set problem. This problem is to determine whether there exists a dominating set (i.e., a subset of vertices such that every vertex in the graph is either in the subset or adjacent to a vertex in the subset) with at most $k$ vertices in a given graph. Given the domination number $k$, the synthetic generator produces an Erdős-Rényi graph with $v$ vertices and a given edge probability $p$ and then transforms the corresponding $k$-Dominating Set problem into a SAT instance. | General: $p = 1 - \left(1 - \binom{v}{k}^{-1/(v-k)}\right)^{1/k}$, Easy dataset: $v \sim \text{Uniform}(5, 15)$, $k \sim \text{Uniform}(2, 3)$, Medium dataset: $v \sim \text{Uniform}(15, 20)$, $k \sim \text{Uniform}(3, 5)$, Hard dataset: $v \sim \text{Uniform}(20, 25)$, $k \sim \text{Uniform}(4, 6)$ | The parameter $p$ is selected based on the paper [40], making the expected number of domination set with size $k$ in the generated graph equals 1. |
| $k$-Vercov | The $k$-Vercov dataset consists of SAT instances that encode the $k$-Vertex Cover problem, i.e., check whether there exists a set of $k$ vertices in a graph such that every edge has at least one endpoint in this set. Given the vertex cover number $k$, the synthetic generator produces a complement graph of an Erdős-Rényi graph with $v$ vertices and a given edge probability $p$ and then converts the corresponding $k$-Vertex Cover problem into a SAT instance. | General: $p = \binom{v}{k}^{-1/\binom{v}{2}}$, Easy dataset: $v \sim \text{Uniform}(5, 15)$, $k \sim \text{Uniform}(3, 5)$, Medium dataset: $v \sim \text{Uniform}(10, 20)$, $k \sim \text{Uniform}(6, 8)$, Hard dataset: $v \sim \text{Uniform}(15, 25)$, $k \sim \text{Uniform}(9, 10)$ | The parameter $p$ is selected based on the relationship between $k$-Vertex Cover and $k$-Clique problems, making the size of the minimum vertex cover in the generated graph around $k$. |

**Statistics.** To provide a comprehensive understanding of our generated datasets, we compute several characteristics across three difficulty levels. These statistics include the average number of variables and clauses, as well as graph measures such as average clustering coefficient (in VIG) and modularity (in VIG, VCG, and LCG). The dataset statistics are summarized in Table 8.

Table 8: Dataset statistics across difficulty levels in G4SATBench.

| Dataset | Easy Difficulty | | | | | | Medium Difficulty | | | | | | Hard Difficulty | | | | | |
|---|---|---|---|---|---|---|---|---|---|---|---|---|---|---|---|---|---|---|
| | #Variables | #Clauses | C.C.(VIG) | Mod.(VIG) | Mod.(VCG) | Mod.(LCG) | #Variables | #Clauses | C.C.(VIG) | Mod.(VIG) | Mod.(VCG) | Mod.(LCG) | #Variables | #Clauses | C.C.(VIG) | Mod.(VIG) | Mod.(VCG) | Mod.(LCG) |
| SR | 25.00 | 148.35 | 0.98 | 0.00 | 0.25 | 0.33 | 118.36 | 646.54 | 0.62 | 0.06 | 0.31 | 0.37 | 299.64 | 1613.86 | 0.32 | 0.09 | 0.32 | 0.37 |
| 3-SAT | 25.05 | 113.69 | 0.72 | 0.06 | 0.36 | 0.46 | 120.00 | 513.14 | 0.27 | 0.16 | 0.43 | 0.51 | 250.44 | 1067.34 | 0.14 | 0.17 | 0.45 | 0.52 |
| CA | 31.66 | 303.48 | 0.65 | 0.19 | 0.73 | 0.73 | 120.27 | 1661.07 | 0.54 | 0.38 | 0.80 | 0.80 | 299.68 | 4195.50 | 0.59 | 0.57 | 0.80 | 0.80 |
| PS | 25.41 | 176.68 | 0.98 | 0.00 | 0.27 | 0.32 | 118.75 | 822.78 | 0.86 | 0.05 | 0.35 | 0.37 | 249.61 | 1728.34 | 0.77 | 0.08 | 0.38 | 0.28 |
| $k$-Clique | 34.85 | 592.89 | 0.90 | 0.03 | 0.45 | 0.49 | 69.56 | 2220.05 | 0.91 | 0.03 | 0.48 | 0.49 | 112.87 | 5543.26 | 0.88 | 0.04 | 0.49 | 0.50 |
| $k$-Domset | 41.90 | 369.40 | 0.70 | 0.26 | 0.47 | 0.53 | 90.64 | 1736.22 | 0.70 | 0.21 | 0.49 | 0.51 | 137.31 | 4032.48 | 0.70 | 0.20 | 0.49 | 0.51 |
| $k$-Vercov | 45.41 | 484.28 | 0.66 | 0.16 | 0.48 | 0.53 | 107.40 | 2634.14 | 0.69 | 0.16 | 0.49 | 0.51 | 190.24 | 8190.94 | 0.69 | 0.16 | 0.50 | 0.51 |

## B  GNN Models

**Message-passing schemes on VCG\*.**  Recall that VCG* incorporates two distinct edge types, G4SATBench employs different functions to execute heterogeneous message-passing in each direction of each edge type. Formally, we define a $d$-dimensional embedding for each variable and clause node, denoted by $h_l$ and $h_c$, respectively. These embeddings are initialized to two learnable vectors $h_v^0$ and $h_c^0$, depending on the node type. At the $k$-th iteration of message passing, these hidden representations are updated as follows:

$$h_c^{(k)} = \text{UPD}\left(\underset{v \in c^+}{\text{AGG}}\left(\left\{\text{MLP}_v^+\left(h_v^{(k-1)}\right)\right\}\right), \underset{v \in c^-}{\text{AGG}}\left(\left\{\text{MLP}_v^-\left(h_v^{(k-1)}\right)\right\}\right), h_c^{(k-1)}\right),$$
$$h_v^{(k)} = \text{UPD}\left(\underset{c \in v^+}{\text{AGG}}\left(\left\{\text{MLP}_c^+\left(h_c^{(k-1)}\right)\right\}\right), \underset{c \in v^-}{\text{AGG}}\left(\left\{\text{MLP}_c^-\left(h_c^{(k-1)}\right)\right\}\right), h_v^{(k-1)}\right), \quad (7)$$

where $c^+$ and $c^-$ denote the sets of variable nodes that occur in the clause $c$ with positive and negative polarity, respectively. Similarly, $v^+$ and $v^-$ denote the sets of clause nodes where variable $v$ occurs in positive and negative form. $\text{MLP}_v^+$, $\text{MLP}_v^-$, $\text{MLP}_c^+$, and $\text{MLP}_c^-$ are four MLPs. $\text{UPD}(\cdot)$ is the update function, and $\text{AGG}(\cdot)$ is the aggregation function.

**GNN baselines.**  Table 9 summarizes the message-passing algorithms of the GNN models used in G4SATBench. We adopt heterogeneous versions of GCN [23], GGNN [27], and GIN [41] on both LCG* and VCG*, while maintaining the original NeuroSAT [34] only on LCG*.

Table 9: Supported GNN models in G4SATBench.

| Graph | Method | Message-passing Algorithm | Notes |
|---|---|---|---|
| LCG* | NeuroSAT | $h_c^{(k)}, s_c^{(k)} = \text{LayerNormLSTM}_1\left(\sum_{l \in \mathcal{N}(c)} \text{MLP}_l\left(h_l^{(k-1)}\right), \left(h_c^{(k-1)}, s_c^{(k-1)}\right)\right),$ $h_l^{(k)}, s_l^{(k)} = \text{LayerNormLSTM}_2\left(\left[\sum_{c \in \mathcal{N}(l)} \text{MLP}_c\left(h_c^{(k-1)}\right), h_{\neg l}^{(k-1)}\right], \left(h_l^{(k-1)}, s_l^{(k-1)}\right)\right)$ | $s_c, s_l$ are the hidden states which are initialized to zero vectors. |
| | GCN | $h_c^{(k)} = \text{Linear}_1\left(\left[\sum_{l \in \mathcal{N}(c)} \frac{\text{MLP}_l\left(h_l^{(k-1)}\right)}{\sqrt{d_l d_c}}, h_c^{(k-1)}\right]\right),$ $h_l^{(k)} = \text{Linear}_2\left(\left[\sum_{c \in \mathcal{N}(l)} \frac{\text{MLP}_c\left(h_c^{(k-1)}\right)}{\sqrt{d_c d_l}}, h_{\neg l}^{(k-1)}, h_l^{(k-1)}\right]\right)$ | $d_c, d_l$ are the degrees of clause node $c$ and literal node $l$ in LCG respectively. |
| | GGNN | $h_c^{(k)} = \text{GRU}_1\left(\sum_{l \in \mathcal{N}(c)}\left(\left\{\text{MLP}_l\left(h_l^{(k-1)}\right)\right\}\right), h_c^{(k-1)}\right),$ $h_l^{(k)} = \text{GRU}_2\left(\left[\sum_{c \in \mathcal{N}(l)} \text{MLP}_c\left(h_c^{(k-1)}\right), h_{\neg l}^{(k-1)}\right], h_l^{(k-1)}\right)$ | |
| | GIN | $h_c^{(k)} = \text{MLP}_1\left(\left[\sum_{l \in \mathcal{N}(c)}\left(\left\{\text{MLP}_l\left(h_l^{(k-1)}\right)\right\}\right), h_c^{(k-1)}\right]\right),$ $h_l^{(k)} = \text{MLP}_2\left(\left[\sum_{c \in \mathcal{N}(l)} \text{MLP}_c\left(h_c^{(k-1)}\right), h_{\neg l}^{(k-1)}, h_l^{(k-1)}\right]\right)$ | |
| VCG* | GCN | $h_c^{(k)} = \text{Linear}_1\left(\left[\sum_{v \in c^+} \frac{\text{MLP}_v^+\left(h_v^{(k-1)}\right)}{\sqrt{d_v d_c}}, \sum_{v \in c^-} \frac{\text{MLP}_v^-\left(h_v^{(k-1)}\right)}{\sqrt{d_v d_c}}, h_c^{(k-1)}\right]\right),$ $h_v^{(k)} = \text{Linear}_2\left(\left[\sum_{c \in v^+} \frac{\text{MLP}_c^+\left(h_c^{(k-1)}\right)}{\sqrt{d_c d_v}}, \sum_{c \in v^-} \frac{\text{MLP}_c^-\left(h_c^{(k-1)}\right)}{\sqrt{d_c d_v}}, h_v^{(k-1)}\right]\right)$ | $d_c, d_v$ are the degrees of clause node $c$ and variable node $v$ in VCG respectively. |
| | GGNN | $h_c^{(k)} = \text{GRU}_1\left(\left[\sum_{v \in c^+} \text{MLP}_v^+\left(h_v^{(k-1)}\right), \sum_{v \in c^-} \text{MLP}_v^-\left(h_v^{(k-1)}\right)\right], h_c^{(k-1)}\right),$ $h_v^{(k)} = \text{GRU}_2\left(\left[\sum_{c \in v^+} \text{MLP}_c^+\left(h_c^{(k-1)}\right), \sum_{c \in v^-} \text{MLP}_c^-\left(h_c^{(k-1)}\right)\right], h_v^{(k-1)}\right)$ | |
| | GIN | $h_c^{(k)} = \text{MLP}_1\left(\left[\sum_{v \in c^+} \text{MLP}_v^+\left(h_v^{(k-1)}\right), \sum_{v \in c^-} \text{MLP}_v^-\left(h_v^{(k-1)}\right), h_c^{(k-1)}\right]\right),$ $h_v^{(k)} = \text{MLP}_2\left(\left[\sum_{c \in v^+} \text{MLP}_c^+\left(h_c^{(k-1)}\right), \sum_{c \in v^-} \text{MLP}_c^-\left(h_c^{(k-1)}\right), h_v^{(k-1)}\right]\right)$ | |

## C  Benchmarking Evaluation

### C.1  Implementation Details

In G4SATBench, we provide the ground truth of satisfiability and satisfying assignments by calling the state-of-the-art modern SAT solver CaDiCaL [13] and generate the truth labels for unsat-core

522 variables by invoking the proof checker DRAT-trim [39]. All neural networks in our study are
523 implemented using PyTorch [31] and PyTorch Geometric [12]. For all GNN models, we set the
524 feature dimension $d$ to 128 and the number of message passing iterations $T$ to 32. The MLPs
525 in the models consist of two hidden layers with the ReLU [29] activation function. To select the
526 optimal hyperparameters for each GNN baseline, we conduct a grid search over several settings.
527 Specifically, we explore different learning rates from $\{10^{-3}, 5 \times 10^{-4}, 10^{-4}, 5 \times 10^{-5}, 10^{-5}\}$,
528 training epochs from $\{50, 100, 200\}$, weight decay values from $\{10^{-6}, 10^{-7}, 10^{-8}, 10^{-9}, 10^{-10}\}$,
529 and gradient clipping norms from $\{0.1, 0.5, 1\}$. We employ Adam [22] as the optimizer and set the
530 batch size to 128, 64, or 32 to fit within the maximum GPU memory (48G). For the parameters $\tau$
531 and $\kappa$ of the unsupervised loss in Equation 4 and Equation 5, we try the default settings ($\tau = t^{-0.4}$
532 and $\kappa = 10$, where $t$ is the global step during training) as the original paper [1] as well as other
533 values ($\tau \in \{0.05, 0.1, 0.2, 0.5\}$, $\kappa \in \{1, 2, 5\}$) and empirically find $\tau = 0.1, \kappa = 1$ yield the best
534 results. Furthermore, it is important to note that we use three different random seeds to benchmark
535 the performance of different GNN models and assess the generalization ability of NeuroSAT and
536 GGNN using one seed for simplicity.

## C.2 Satiafiability Prediction

538 **Evaluation across different difficulty levels.** The complete results of NeuroSAT and GGNN across
539 different difficulty levels are presented in Figure 6. Consistent with the findings on the SR and 3-SAT
540 datasets, both GNN models exhibit limited generalization ability to larger instances beyond their
541 training data, while displaying relatively better performance on smaller instances. This observation
542 suggests that training these models on more challenging instances could potentially enhance their
543 generalization ability and improve their performance on larger instances.

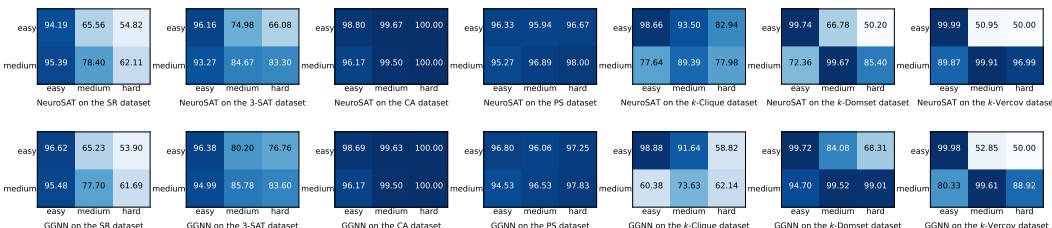

Figure 6: Results across different difficulty levels. The x-axis denotes testing datasets and the y-axis denotes training datasets.

544 **Evaluation with different message passing iterations.** To investigate the impact of message-
545 passing iterations on the performance of GNN models during training and testing, we conducted
546 experiments with varying iteration values. Figure 7 presents the results of NeuroSAT and GGNN
547 trained and evaluated with different message passing iterations. Remarkably, using a training iteration
548 value of 32 consistently yielded the best performance for both models. Conversely, employing too
549 small or too large iteration values during training resulted in decreased performance. Furthermore, the
550 models trained with 32 iterations also demonstrated good generalization ability to testing iterations 16
551 and 64. These findings emphasize the critical importance of selecting an appropriate message-passing
552 iteration to ensure optimal learning and reasoning within GNN models.

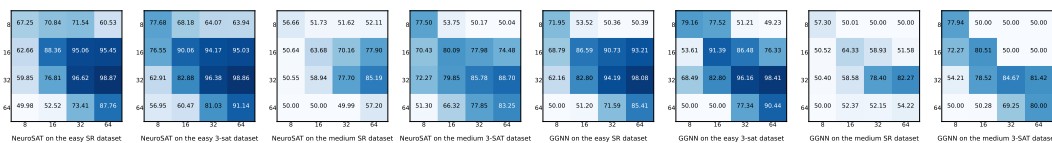

Figure 7: Results across different message passing iterations $T$. The x-axis denotes testing iterations and the y-axis denotes training iterations.

## C.3 Satisfying Assignment Prediction

**Evaluation with different datasets.** Figure 8 illustrates the performance of NeuroSAT across different datasets. For easy datasets, we observe that NeuroSAT demonstrates a strong generalization ability to other datasets when trained on the SR, 3-SAT, CA, and PS datasets. However, when trained on the $k$-Clique, $k$-Domset, and $k$-Vercov datasets, which involve specific graph structures inherent to their combinatorial problems, NeuroSAT struggles to generalize effectively. This observation indicates that the GNN model may overfit to leverage specific graph features associated with these combinatorial datasets, without developing a generalized solving strategy that can be applied to other problem domains for satisfying assignment prediction. For medium datasets, NeuroSAT also faces challenges in generalization, as its performance is relatively limited. This can be attributed to the difficulty of these datasets, where finding satisfying assignments is much harder than easy datasets.

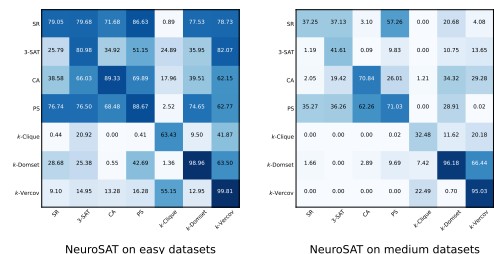

Figure 8: Results of NeuroSAT across different datasets (with $UNS_2$ as the training loss). The x-axis denotes testing datasets and the y-axis denotes training datasets.

**Evaluation across different difficulty levels.** The performance of NeuroSAT across different difficulty levels is shown in Figure 9. Notably, training on medium datasets yields superior generalization performance compared to training on easy datasets. This suggests that training on more challenging instances can enhance the model's ability to generalize to a wider range of problem complexities.

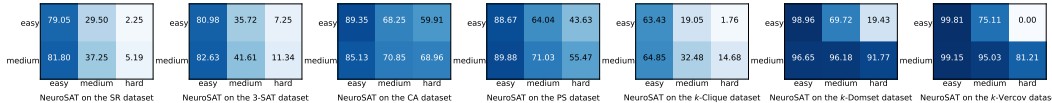

Figure 9: Results of NeuroSAT across different difficulty levels (with $UNS_2$ as the training loss). The x-axis denotes testing datasets and the y-axis denotes training datasets.

**Evaluation with different inference algorithms.** Figure 10 illustrates the results of NeuroSAT using various decoding algorithms (with $UNS_2$ as the training loss). Surprisingly, all three decoding algorithms demonstrate remarkably similar performances across all datasets. This observation indicates that utilizing the standard readout after message passing is sufficient for predicting a satisfying assignment. Also, the GNN model has successfully learned to identify potential satisfying assignments within the latent space, which can be extracted by clustering the literal embeddings.

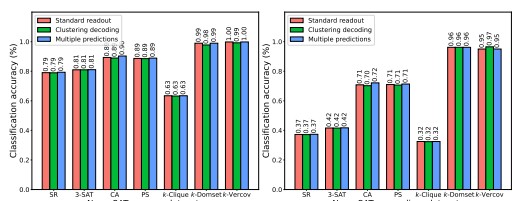

Figure 10: Results of NeuroSAT with different inference algorithms.

**Evaluation with unsatisfiable training instances.** Following previous works [1, 2, 30], our evaluation of GNN models focuses solely on satisfiable instances. However, in practical scenarios, the satisfiability of instances may not be known before training. To address this gap, we explore the effectiveness of training NeuroSAT using the unsupervised loss $UNS_2$ on noisy datasets that contain unsatisfiable instances. Table 10 presents the results of NeuroSAT when trained on such datasets, where 50% of the instances are unsatisfiable. Interestingly, incorporating unsatisfiable instances for training does not significantly affect the performance of the GNN model. This finding highlights the potential utility of training GNN models using $UNS_2$ loss on new datasets, irrespective of any prior knowledge regarding their satisfiability.

Table 10: Results of NeuroSAT when trained on noisy datasets. Values in parentheses indicate the performance difference compared to the model trained without unsatisfiable instances. The $k$-Clique dataset is excluded as NeuroSAT fails during training.

| | Easy Datasets | | | | | | Medium Datasets | | | | |
|---|---|---|---|---|---|---|---|---|---|---|---|
| | SR | 3-SAT | CA | PS | $k$-Domset | $k$-Vercov | SR | 3-SAT | CA | PS | $k$-Domset | $k$-Vercov |
| | 0.7884 | 0.8048 | 0.8701 | 0.8866 | 0.9800 | 0.9524 | 0.3721 | 0.4175 | 0.7649 | 0.7252 | 0.9493 | 0.9618 |
| | (-0.95) | (-0.11) | (-2.33) | (-0.13) | (-0.85) | (-4.49) | (-0.04) | (+0.14) | (+5.64) | (+1.46) | (-1.25) | (+0.19) |

### C.4 Unsat-core Variable Prediction

**Evaluation across different datasets.** Figure 11 shows the generalization results across different datasets. NeuroSAT and GGNN demonstrate good generalization performance to datasets that are different from their training data, except for the CA dataset. This discrepancy can be attributed to the specific characteristics of the CA dataset, where the number of unsat-core variables is significantly smaller compared to the number of variables not in the unsat core. In contrast, other datasets exhibit a different distribution, where the number of variables in the unsat core is much larger. This variation in distribution presents a challenge for the models' generalization ability in the case of the CA dataset.

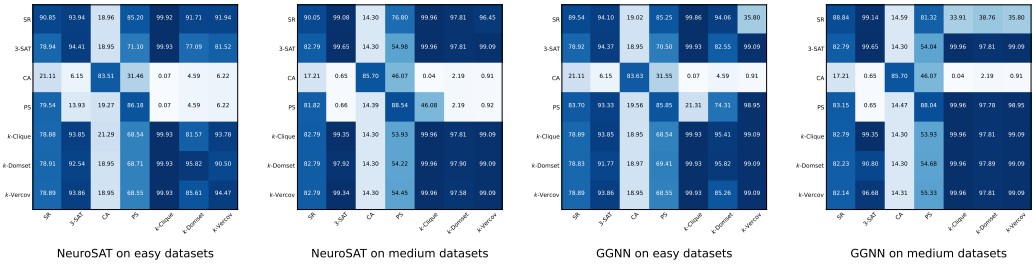

Figure 11: Results across different datasets. The x-axis denotes testing datasets and the y-axis denotes training datasets.

**Evaluation across different difficulty levels.** The results across different difficulty levels are presented in Figure 12. Remarkably, both NeuroSAT and GGNN exhibit a strong generalization ability when trained on easy or medium datasets. This suggests that GNN models can effectively learn and generalize from the characteristics and patterns present in these datasets, enabling them to perform well on a wide range of problem complexities.

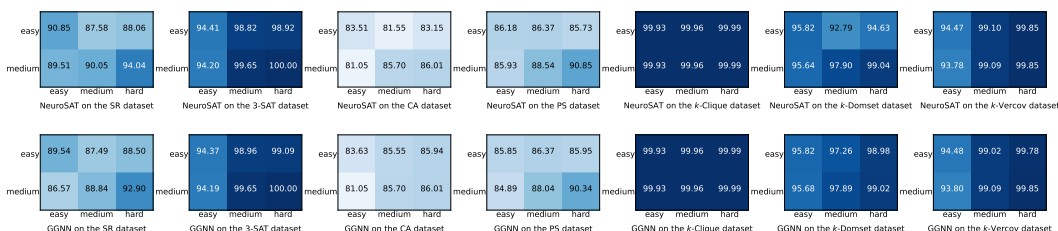

Figure 12: Results across different difficulty levels. The x-axis denotes testing datasets and the y-axis denotes training datasets.

## D  Advancing Evaluation

**Implementation details.** To create the augmented datasets, we leverage CaDiCaL [13] to generate a DART proof [39] for each SAT instance, which tracks the clause learning procedure and records all the learned clauses during the solving process. These learned clauses are then added to each instance, with a maximum limit of 1,000 clauses. For experiments on augmented datasets, we keep all training settings identical to those used for the original datasets.

For contrastive pretraining experiments, we treat each original formula and its augmented counterpart as a positive pair and all other instances in a mini-batch as negative pairs. We use an MLP projection to map the graph embedding $z_i$ of each formula to $m_i$ and employ the SimCLR's contrastive loss [8], where the loss function for a positive pair of examples $(i, j)$ in a mini-batch of size $2N$ is defined as:

$$\mathcal{L}_{i,j} = - \log \frac{\exp(\text{sim}(m_i, m_j)/\tau)}{\sum_{k=1}^{2N} \mathbb{1}_{[k \neq i]} \exp(\text{sim}(m_i, m_k)/\tau)}. \tag{8}$$

Here, $\mathbb{1}_{[k \neq i]}$ is an indicator function that evaluates to 1 if $k \neq i$, $\tau$ is a temperature parameter, and $\text{sim}(\cdot, \cdot)$ is the similarity function defined as $\text{sim}(m_i, m_j) = m_i^\top m_j / \|m_i\| \|m_j\|$. The final loss is the average over all positive pairs. In our experiments, we set the temperature parameter to 0.5 and utilize a learning rate of $10^{-4}$ with a weight decay of $10^{-8}$. The pretraining process is performed for a total of 100 epochs. Once the pretraining is completed, we only keep the GNN model and remove the projection head for downstream tasks.

For experiments involving random initialization, we utilize Kaiming Initialization [18] to initialize all literal/variable and clause embeddings during both training and testing. For the predicted assignments, we utilize 2-clustering decoding to construct two possible assignment predictions for NeuroSAT* at each iteration. When calculating the number of flipped variables and the number of unsatisfiable clauses for NeuroSAT*, we only consider the better assignment prediction of the two at each iteration, which is the one that satisfies more clauses. All other experimental settings remain the same as in the benchmarking evaluation.