# OpenReview forum: "G4SATBench: Benchmarking and Advancing SAT Solving with Graph Neural Networks"
_NeurIPS.cc/2023/Track/Datasets_and_Benchmarks — Submitted to NeurIPS 2023 Datasets and Benchmarks_

### Official Review · Reviewer_yiUC · 2023-07-21
**This work presents G4SATBench, which offers curated synthetic SAT datasets sourced from various domains and difficulty levels and benchmarks a wide range of GNN-based SAT solvers under diverse settings.**

**Rating:** 7
**Confidence:** 4
**Clarity:** The paper is well-written and easy to…

**Strengths:**

1. The authors try to answer the question of how well GNNs can solve SAT. The variety of learning objectives and usage scenarios employed in existing work, makes it difficult to evaluate different methods fairly and comprehensively.
2. The authors re-implement four GNN-based SAT solvers with unified interfaces and configuration settings in existing works. That helps to provide baselines for GNN-based SAT solvers.

**Additional Feedback:**

Figures 3, 4, and 5 are difficult to read and recognize. To make them clear, please display them in a larger font.

**Correctness:**

Most of the claims made in the submission are correct.
The framework of G4SATBench offers curated synthetic SAT datasets sourced from various domains and difficulty levels and benchmarks a wide range of GNN-based SAT solvers under diverse settings.
The experiment design is appropriate and performed correctly.

**Documentation:**

Accessing the URL offered by the authors, there are some sufficient details about the collection, organization, availability, hosting, licensing, and maintenance of datasets.

**Ethics:**

There is no ethical concern.

**Limitations:**

The authors did good work on GNN-based SAT solvers with synthetic SAT datasets. The following limitations can be explored in the future:
1. G4SATBench primarily focuses on evaluating standalone neural SAT solvers, excluding the exploration of neural-guided SAT solvers that integrate GNNs with search-based SAT solvers.
2. The instances included in G4SATBench are relatively small compared to most practical instances found in real-world applications.

**Opportunities For Improvement:**

1. In the part of Graph constructions, the authors should give more explanation about the slight modifications of LCG* and VCG* compared with LCG and VCG (why not sophisticated GNN or other fully connected constructions), especially for the differences in both constructions and impact on baseline results.
2. In Section 5.1, based on the observation, the performance of all GNN models is strong on easy datasets and four medium datasets, except the medium SR, 3-SAT, and $k$-Clique datasets, which incompletely matches the author's description. Please explain why the performance is lower in medium SR, 3-SAT, and $k$-Clique datasets than that of easy datasets.
3. The GNN-based SAT solvers collected in G4SATBench are not enough and should be increased in the future.

**Relation To Prior Work:**

The authors discussed how this work differs from previous contributions.
The existing SAT datasets are not particularly amenable for GNNs to learn from, given their relatively modest scale or overly extensive instances.  The existing synthetic SAT datasets are generated from a few domains or non-evaluation. To address this issue, the authors provide a G4SATBench framework that contains 7 datasets from 3 domains and baseline results from different levels.

**Summary And Contributions:**

1. The benchmark is built on a diverse set of synthetic CNF generators, that consists of 7 different datasets from 3 domains.
2. G4SATBench offers easy and medium levels for all datasets to enable fine-grained analyses.
3. G4SATBench framework evaluates different GNN models in SAT solving with various prediction tasks, training objectives, and inference algorithms.
4. The authors find that GNNs tend to develop a solving heuristic similar to the greedy local search to find a satisfying assignment but fail to effectively learn the backtracking heuristic in the latent space.

---

> ### Author Response · Authors · 2023-08-12
>
> Thank you for your valuable feedback on our paper! In the following, we hope to address each of your concerns:
>
> > More explanation about the slight modifications of LCG* and VCG* compared with LCG and VCG.
>
> In the LCG, there's no edge linking the positive and negative literals of the same variable. If GNNs are directly applied to the unmodified LCG, it would omit this crucial connection, thereby obfuscating which positive/negative literals correspond to the same variable. Similarly, for VCG, utilizing a single edge type would render us unable to differentiate variable polarities within a clause. Thus, these modifications are applied to the original ones to distinguish the polarity of the same variable. We have detailed these modifications further in our paper's revision for clarity.
>
> > Why not utilize sophisticated GNN or other fully connected constructions?
>
> While LCG* and VCG* are the predominant graph constructions for GNN-based SAT solvers, we recognize the potential of other encodings like the And-Inverter-Graph (AIG) for SAT instances *not in CNF*. However, such representations are typically specialized to specific applications (like CircuitSAT) and not designed for general purposes. Given this specialization, we've chosen to keep them outside the scope of the current G4SATBench, though we remain optimistic about exploring such sophisticated graph constructions and GNN models in future research.
>
> > Why the performance is lower in medium SR, 3-SAT, and $k$-Clique datasets than that of easy datasets?
>
> We want to clarify that our categorization into "easy," "medium," and "hard" datasets is primarily based on instance size (i.e., the number of variables and clauses in an instance). However, the inherent characteristics and distinct logical structures of each dataset can lead to variations in actual difficulty. In the SR dataset, the *minimal* difference (i.e., *just one* different literal) between each satisfiable/unsatisfiable pair challenges GNN models' discriminative capabilities for satisfiability prediction. For the 3-SAT dataset, our hyperparameter choices ensure a balanced distribution of satisfiable/unsatisfiable instances around the *phase transition*, posing a distinct challenge. Similarly, our $k$-Clique datasets have instances with expected $k$-Cliques equal to 1. These intricacies contribute to the different performance in classification accuracy observed in these datasets.
>
> > The GNN-based SAT solvers collected in G4SATBench are not enough and should be increased in the future.
>
> While we include a range of GNN-based SAT solvers mainly on the LCG/VCG representations in G4SATBench, we anticipate future studies to introduce other GNN-based solvers with alternative graph encodings or message-passing schemes. We have provided detailed instructions/comments on how to extend our work in our GitHub repository.
>
>
> > Figures 3, 4, and 5 are difficult to read and recognize.
>
> We have reorganized the experiment section and increased the font size in the figures as well.

---

### Official Review · Reviewer_upz8 · 2023-07-21
**Good paper benchmarking GNN SAT solvers**

**Rating:** 9
**Confidence:** 3

**Strengths:**

1. The paper and code and are clearly written and make it easy for future authors to compare new methods.
2. The authors gain use their benchmarks to obtain new insights that could guide future research in the area.
3. The generalization experiments (and failure of GNNs to generalize well) provide a strong argument for the mixture of tasks that the authors have chosen.
4. The paper provides valuable insight into how GNN based SAT solvers work

**Additional Feedback:**

n/a

**Clarity:**

Yes the paper is generally well written and easy to follow.

A few issues I ran into:
- Some of the acronyms don't seem to be fully explained e.g. I could not find the meaning of SR in the paper.

One complaint is that Figure 3 is a bit hard to read on paper because the font size is so small.  The tables also seem in Fig 3 also seem to have a typo NeoruSAT -> NeuroSAT

**Correctness:**

The datasets used are constructed in a sound way and care is taken to avoid trivial problem instances. The evaluation methods also seem appropriate and performed correctly.

**Documentation:**

Yes the code is available on GitHub

**Limitations:**

Yes - the primary limitations as I see it are that the benchmark only addresses pure GNN SAT solvers without addressing GNN guided SAT solvers and that the instances are still relatively small. The authors discuss this in the appendix.

**Opportunities For Improvement:**

1.  A bit more detail about the datasets used would be helpful (e.g. the comment about SR below)

**Relation To Prior Work:**

Yes the authors spend time motivating the need for the type of dataset that they are constructing and give some explicit differences to otthers.

**Summary And Contributions:**

1. The paper collects a benchmark suite of a variety of SAT problems for comparing GNN based SAT solvers. These include both synthetic datasets and 'pseudo-industrial' datasets.
2. The paper also implements a wide variety of GNN architectures which have been previously published as SAT solvers in a single framework and compares them. Including different graph representations of SAT problems, losses, and readout algorithms.
3. The authors evaluate the various algorithms by measuring overall performance for Satisfiability prediction, assignment prediction,  and unsat-core prediction.  Including measuring generalizability across different instance types.
4. Finally the authors gain some insights into work their solvers operate by augmenting them with different commonly used heuristics.

---

> ### Author Response · Authors · 2023-08-12
>
> Thanks a lot for your positive review and valuable feedback! We hope to address your concerns and questions about our paper as follows:
>
> >  The meaning of SR in the paper.
>
> The SR dataset contains pairs of satisfiable/unsatisfiable instances, where there is *only one different literal* between each pair. This setting poses a challenge for GNN models to classify the *satisfiability* between each pair, since there is only one differing edge between two LCG*/VCG* representations but the label for them is different. Furthermore, this design choice may also prevent GNN to reply on the spurious feature (e.g., variable/clause ratio, degree of each variable, etc.) for satisfiability prediction.
>
> > Figure 3 is a bit hard to read on paper because the font size is so small. The tables also seem in Fig 3 also seem to have a typo NeoruSAT -> NeuroSAT.
>
> We have reorganized the experiment section and increased the font size in the figures. Moreover, we have also elaborated on our paper to fix typos in the revision.

---

### Official Review · Reviewer_CeEF · 2023-07-21

**Rating:** 6
**Confidence:** 3
**Correctness:** Correct as far as I can tell.
**Clarity:** Yes.

**Strengths:**

The paper is well-written, addresses and interesting problem, and makes a good contribution to the literature. To the best of my knowledge, there is no standard benchmark dataset for GNN-based SAT solving, which, as the authors point out, makes results in the literature unnecessarily difficult to compare. The proposed benchmark addresses this issue.

**Additional Feedback:**

Overall the paper is very nice, but I am concerned with the lack of documentation in the github repository.

**Documentation:**

The dataset documentation could be improved. There is no description of data formats etc in the github repository, only a short readme. The majority of code seems to have no comments at all. It is unlikely that somebody else would be able to modify or extend the benchmark without substantial additional work.

**Opportunities For Improvement:**

My main complaint is that the font in tables and figures is too small and almost unreadable. Please increase the font size. Please also add some highlighting to the tabular results (e.g. the best for each dataset) or present in the form of a graph, as a lot of numbers are not intuitive to understand.

**Relation To Prior Work:**

Yes.

**Summary And Contributions:**

The paper proposes a benchmark for graph-neural-network-based SAT solving. The authors describe the benchmark set and give some empirical results.

---

> ### Author Response · Authors · 2023-08-12
>
> Thank you for the positive feedback and suggestions! Here's how we have addressed the concerns you have mentioned:
>
> > The font in tables and figures is too small and almost unreadable.
>
> We have reorganized the experiment section, increased the font size in figures, and also highlighted the numbers in tables. We believe these adjustments ensure improved readability and better clarity for our readers.
>
> > The dataset documentation could be improved.
>
> We have added more details to the README.md file in our GitHub repository, including the folder structure, data format, and guidelines for extending our work. Additionally, we have incorporated thorough comments within nearly all of our files, ensuring that researchers can easily understand, utilize, and build upon our approach.

---

> > ### Comment · Reviewer_CeEF · 2023-08-24
> >
> > Thank you for the changes.

---

### Official Review · Reviewer_oRgT · 2023-07-23
**Review for G4SATBench**

**Rating:** 7
**Confidence:** 3
**Clarity:** Yes, the writing is clear.

**Strengths:**

The curated dataset in this paper is one of the first benchmark datasets that aims to asses GNN-based SAT solvers. The is a clear contribution to the set of such problems in that it provides a general dataset that can be used to test different prediction tasks SAT solvers aim to perform.

**Additional Feedback:**

There's no additional feedback.

**Correctness:**

The dataset is constructed in a sound way. And the evaluation methods and experiment design are appropriate and performed correctly.

There are no notable errors.

**Documentation:**

It is well documented, and a URL for the GitHub page was provided.

**Ethics:**

There are no ethical concerns.

**Limitations:**

This is not aligned with the main emphasis of the paper to create unified interfaces and configuration settings, but one of the major limitations highlighted in the review is that the GNN models tend to develop a solving heuristic similar to the greedy local search but fail to learn the backtracking heuristic in the latent space effectively. This could be the reason to consider model-specific optimization techniques to leverage the capabilities of each individual GNN model fully.

**Opportunities For Improvement:**

As the authors have mentioned in Appendix E, the generalization ability of the GNN-based SAT solvers across different data distributions and instance sizes might be limited. How the solvers perform on real-world problems outside the curated datasets could be another area of limitation.

The research could benefit from an even wider variety of datasets, perhaps incorporating more real-world scenarios or instances from more diverse fields.

As mentioned in the limitations, while the unified interface allows for the fair comparison of different GNN models, it may not account for all model-specific optimization techniques. This might influence the comparison results.


**Relation To Prior Work:**

 It re-implements various GNN-based SAT solvers, ones developed in previous studies, with unified interfaces and configuration settings. This introduces a general evaluation protocol for a more comprehensive and equitable comparison of different models, which could address potential inconsistencies or gaps in the methodologies of past research.

The introduction of baseline results is another significant contribution that provides a solid foundation for future work. These baselines can be used for comparisons with prior and future research, offering a reference point to measure progress in the field.

**Summary And Contributions:**

This study delivers an exhaustive assessment of GNN-based SAT solvers. The authors have compiled a comprehensive benchmark of diverse SAT datasets for robust analysis. They've also re-implemented various GNN-based SAT solvers with unified interfaces, facilitating fair comparisons between different models. Baseline results are presented, offering insights into GNN model performances under different conditions. In a series of experiments, GNNs were found to be proficient in adopting a greedy local search heuristic, but less effective in learning backtracking heuristic. The study serves as an invaluable foundation for future research, albeit underlining certain challenges in the learning capabilities of GNNs.

---

> ### Author Response · Authors · 2023-08-12
>
> Thank you for your valuable feedback! In the following, please let us address the concerns you've pointed out.
>
> >  The generalization ability of the GNN-based SAT solvers across different data distributions and instance sizes might be limited.
>
> We acknowledge that the generalization ability of GNN-based SAT solvers across different datasets might be limited when trained on a single dataset. Moreover, our empirical results show that their generalization capability is also influenced by the inherent logical structure and complexity of the training data. For instance, training on more challenging instances, like those in SR datasets, could lead to better generalization results than other datasets. We believe training GNN models on *multiple harder* datasets might enhance the generalization ability further and leave this exploration as future work.
>
> > Incorporating more real-world scenarios or instances from more diverse fields.
> Response to Reviewer X5tc
>
> It is worth noting, as stated in our paper, that current real-world instances are not particularly amenable for GNNs to learn from. This is attributed to their relatively modest scale (often fewer than 100 instances per specific domain) or their overwhelming size (sometimes surpassing 10 million variables and clauses). Nonetheless, we acknowledge that GNN-based SAT solvers could benefit a lot from training on large-scale and aptly-sized real-world SAT instances. We expect that future research will build such datasets to improve the learning ability of GNN-based SAT solvers and generalize them to real-world applications.
>
> > Considering model-specific optimization techniques.
>
> In our current G4SATBench, our main focuses are the widely-used graph constructions, GNN models, prediction tasks, and learning objectives for SAT solving. While we recognize the potential of other encodings like the And-Inverter-Graph (AIG) and sophisticated GNN models for SAT instances *not in CNF*, such designs are typically specialized to specific applications (like CircuitSAT) and not designed for general purposes. Given this specialization, we've chosen to keep them outside the scope of the current G4SATBench, though we remain optimistic about exploring such model-specific optimization techniques in future research.

---

### Author Response · Authors · 2023-08-23

We genuinely appreciate all the reviewers for their helpful feedback on our paper.

Based on the suggestions, we have addressed the typos, restructured the experimental section, and adjusted the font size in our figures for better clarity. Additionally, we have enhanced both the documentation and code in our GitHub repository. We believe these modifications have improved the readability and quality of our work.

Should there be any more questions or points of clarification needed, we are keen to engage in further discussions!

---

### Decision · Program_Chairs · 2023-09-22

**Decision:**

Reject

**Comment:**

Let me start by saying that this is a difficult decision, and at this stage I am not 100% sure what I should recommend. What is clear that the reviews were all positive, and based on the review scores alone, this paper should be accepted. I also like the idea of a SAT benchmark.

Unfortunately, in my opinion, this paper has a fundamental flaw. I tried to discuss this with the reviewers, but the feedback I got was very limited.

Here is the flaw: I don't think that a benchmark should focus on GNNs only. This is like proposing an MNIST-benchmark but restricting allowed methods to decision forests (or whatever your favorite method is). I think good benchmarks (accepted to NeurIPS) need to be agnostic regarding the method.

In the case of SAT, the very active SAT solving community already has strong standards regarding benchmarks, for instance the yearly SAT Solving Competition. This paper does not talk about classical SAT solving approaches, and does not even mention anything in the long history of SAT solving and SAT solving benchmarks. While GNNs may be a good tool to solve SAT problems, I am not convinced that they can hold a candle to the very elaborate SAT heuristics. After all, there are SAT solving conferences (e.g. http://satisfiability.org/SAT23/, but also other conferences) which have already 25 years of history! I would even argue that SAT solving is the best benchmarked problem in computer science, and I am really worried that we just accept a paper that ignores the whole tradition of SAT solving benchmarks. This is very risky, as there are hundreds of professors out there who have done nothing else but SAT solving benchmarks in their career, and if NeurIPS comes along with a SAT solving benchmark ignoring all their work, they will be upset.

On the other hand, even if we wanted to accept a GNN-only SAT benchmark (for whatever reason), then I find this paper very underwhelming as well. They test their benchmark with 3 old GNN architectures, and completely ignore any of the GNN extensions, which have been a main thread in GNN research in the past few years. So if this was a GNN-only SAT benchmark (ignoring the 25 years of history), even then I would be underwhelmed by the paper.

Meta-comment: I understand that this metareview is totally against the spirit of metareviews, and I'm happy to update it, but I'm very unhappy with the reviews of this paper (and unfortunately also with many of the reviews of the other papers that I area chaired). My other metareviews will be standard.